# Expression Quantitative Trait Methylation Analysis Identifies Whole Blood Molecular Footprint in Fetal Alcohol Spectrum Disorder (FASD)

**DOI:** 10.3390/ijms24076601

**Published:** 2023-04-01

**Authors:** Izabela M. Krzyzewska, Peter Lauffer, Adri N. Mul, Liselot van der Laan, Andrew Y. F. Li Yim, Jan Maarten Cobben, Jacek Niklinski, Monika A. Chomczyk, Robert Smigiel, Marcel M. A. M. Mannens, Peter Henneman

**Affiliations:** 1Genome Diagnostics Laboratory, Department of Human Genetics, Amsterdam University Medical Centers, University of Amsterdam, Meibergdreef 9, 1105 AZ Amsterdam, The Netherlands; 2Department of Clinical Genetics, Amsterdam University Medical Centers, University of Amsterdam, Meibergdreef 9, 1105 AZ Amsterdam, The Netherlands; 3Department of Pediatric Endocrinology and Faculty of Medicine, Northwest Thames Regional Genetics NHS, Imperial College, London SW7 2BX, UK; 4Department of Molecular Biology, Medical University of Bialystok, Jana Kilińskiego 1, 15-089 Białystok, Poland; 5Department of Genetics, Medical University of Wroclaw, Wybrzeże Ludwika Pasteura 1, 50-367 Wrocław, Poland

**Keywords:** FASD, fetal alcohol spectrum disorder, gene expression, DNA methylation, eQTM

## Abstract

Fetal alcohol spectrum disorder (FASD) encompasses neurodevelopmental disabilities and physical birth defects associated with prenatal alcohol exposure. Previously, we attempted to identify epigenetic biomarkers for FASD by investigating the genome-wide DNA methylation (DNAm) profiles of individuals with FASD compared to healthy controls. In this study, we generated additional gene expression profiles in a subset of our previous FASD cohort, encompassing the most severely affected individuals, to examine the functional integrative effects of altered DNAm status on gene expression. We identified six differentially methylated regions (annotated to the *SEC61G*, *REEP3*, *ZNF577*, *HNRNPF*, *MSC,* and *SDHAF1* genes) associated with changes in gene expression (*p*-value < 0.05). To the best of our knowledge, this study is the first to assess whole blood gene expression and DNAm-gene expression associations in FASD. Our results present novel insights into the molecular footprint of FASD in whole blood and opens opportunities for future research into multi-omics biomarkers for the diagnosis of FASD.

## 1. Introduction

Fetal alcohol syndrome (FAS) is a congenital syndrome characterized by neurodevelopmental disabilities and physical characteristics associated with prenatal alcohol exposure (PAE). FAS is the most severe entity within the fetal alcohol spectrum disorders (FASD), and its clinical diagnosis is established with the Four-Digit Diagnostic Code when sufficient severity in four key diagnostic areas is present: PAE, growth failure, FAS-specific facial abnormalities, and neurological features such as intellectual impairment [1]. Individuals who do not display all characteristics may be diagnosed with partial FAS (pFAS), alcohol-related neurodevelopmental disorder (ARND), or alcohol-related birth defects (ARBD).

Diagnosing FASD is crucial for providing the best possible support to children and parents, preventing further alcohol-exposed pregnancies, and breaking the intergenerational cycle of FASD. However, the diagnosis of FASD remains a major clinical challenge for three main reasons [2]: (1) signs and symptoms of FASD considerably overlap with other congenital neurodevelopmental disorders; (2) often, there is limited or unclear information about alcohol consumption during pregnancy; and, particularly, (3) the absence of a definitive diagnostic marker for FASD. Consequently, many individuals with FASD are misdiagnosed or undiagnosed [3].

Previously, we attempted to identify an epigenetic diagnostic marker for FASD by investigating the genome-wide DNA methylation (DNAm) profiles of individuals with FASD compared to healthy controls [4]. Four genomic regions with an altered DNAm status were detected. The identified loci were located within or nearby the *GLI2, TNFRSF19*, *DNTA*, and *NECAB3* genes, which have molecular functions related to symptoms of FASD. There is still uncertainty, however, as to whether and how these DNAm alterations are involved functionally in the pathophysiological mechanism of FASD. There is a need for a more comprehensive description of FASD DNAm associations and their downstream functional effects to unravel their role in FASD pathogenesis. Therefore, the main goal of this study was to further characterize the functional molecular footprint of FASD by assessing the (functional) correlation between DNAm changes and gene expression.

DNAm is a dynamic process of reversible changes to DNA, namely, the addition of a methyl group to cytosine at cytosine-guanine dinucleotides (CpGs), which may affect the level of gene expression in several ways. For example, DNA hypermethylation of gene promoters may block the transcriptional machinery, while hypermethylation of gene bodies or annotated regulatory enhancer sites may result in the increased expression of a gene [5]. It is not known whether the previously identified FASD DNAm associations lead to gene expression changes. Here, we generated additional gene expression profiles in a subset of our previous FASD cohort, encompassing the most severely affected individuals [4], to examine the functional integrative effects of altered DNAm status on gene expression—a so-called expression quantitative trait methylation (eQTM) analysis. Our results shed light on the underlying molecular and biological processes relating to FASD DNAm alterations.

## 2. Results

### 2.1. Study Cohort

This study used samples and data previously collected by Cobben et al. for investigating the DNAm signature of individuals with FASD compared to healthy individuals [4]. A selection criterion was applied to the original FASD cohort, decreasing sample heterogeneity, aiming to find the most robust molecular associations: the most severely affected FASD-affected individuals, with ≥three points on the Four-Digit score, were included in the analyses. A total of 12 of 46 individuals with FASD met the selection criterion. RNA was available for 51 of 92 non-PAE controls, who were subsequently included. The mean age in years (±standard deviation) of individuals with FASD was 6.95 (±3.68), and that of controls was 13.20 (±2.94). In total, 7/12 (58.3%) of individuals with FASD were male, and 33/51 (64.7%) of controls were. All selected participants were of Polish descent.

### 2.2. DNA Methylation Data

Blood DNAm HumanMethylation450K (HM450K) data of the 12 individuals with FASD and 51 healthy controls passed all MethylAid quality control checks (default settings) [6]. Correlation plots of the first eight principal components (PCs) with metadata (sex, age, and relative blood cell type counts) and technical data (array position and slide) showed that subject heterogeneity considerably affected DNAm data (Appendix A). Thus, metadata were included as variables in subsequent analyses of DNAm and gene expression data. Technical data marginally influenced DNAm data (Appendix A). Principal component analysis (PCA) did not indicate unexpected separate clustering of samples (Appendix A). The variance explained by PC 1–8 is given in Appendix A.

### 2.3. Epigenome-Wide Loci Associated with FASD

Differential methylation analysis revealed 179 differentially methylated positions (DMPs; CpGs with a significant methylation level difference between FASD and healthy controls) associated with FASD (Appendix A). DMPs are visualized in a volcano plot in Appendix A. A total of 51 DMPs were hypomethylated (methylation levels were lower in individuals with FASD compared to healthy controls), and 128 DMPs were hypermethylated. According to the UCSC annotation, 77 DMPs were associated with the promoter region of a gene (defined as CpG annotation to 5′UTR, the 1500 bp range upstream the transcription start site, or the first exon of a gene) (Appendix A). Since the studied cohort represents a subset of a larger cohort [4], we conducted a sensitivity analysis of the current epigenome-wide association study (EWAS) by evaluating the overlap of DMPs with the previously reported (full cohort) EWAS, employing the same statistical method (lmfit). Of the 179 DMPs, 88 (49.2%) were also significant in the larger EWAS, while 91 CpGs did not reach genome-wide significance or were not present in the top 2000 CpGs (Appendix A) [4]. Notably, the top 10 DMPs of the current EWAS were also significantly associated in the previous EWAS (Appendix A).

Twenty-one differentially methylated regions (DMRs; regions spanning at least two CpGs, with a significant methylation level difference between FASD and healthy controls) were identified, of which five were hypomethylated and sixteen were hypermethylated (Appendix A). Fifteen DMRs could be annotated to a coding gene, while six were annotated to intergenic regions or microRNAs. Five DMRs from the previous (larger) EWAS were replicated [4]. Overall, these results indicate considerable overlap between both EWAS analyses.

### 2.4. Absent Differential Gene Expression in Whole Blood-Derived RNA of Individuals with FASD

Next, RNA extracted from whole blood samples was sequenced. The RIN (RNA integrity number) was sufficient (RIN ≥ 7.0) in 57 samples. Six samples had subpar RIN values; however, quality assessment with FastQC did not yield any inconsistencies. At least 25 million reads were generated for each sample, which were mapped to the human reference transcriptome and quantified. After filtering steps, selecting for expressed genes, 17,054 genes remained for the analysis, approximately 82% of possible genes. An explorative PCA (using the count data of remaining genes) did not indicate unexpected separate clustering of samples (Appendix A). Surprisingly, differential gene expression analysis (correcting for batch effects and metadata) did not identify genes with a false discovery rate (FDR) < 0.05. A total of 631 genes had a nominal *p*-value of < 0.05 (Appendix A).

### 2.5. Weighted Correlation Network Analysis Identified Clusters of Genes Associated with FASD

To explore biological pathways affected in FASD, we carried out a weighted correlation network analysis (WGCNA), exposing the underlying organization of FASD transcriptomics based on gene co-expression networks. RNA sequencing (RNA-seq) data were filtered for lowly expressed genes and residualized to account for batch effects (age, gender, and (DNAm) estimated blood cell type distribution). We identified several correlation patterns among genes across our RNA-seq data, which could be summarized in 10 modules (Appendix A). One of these modules showed a significant negative correlation with FASD (module brown: *r* = −0.29, *p*-value = 0.02). Gene ontology (GO) gene set overrepresentation analysis (ORA) of the genes in module brown (*n* = 1053) showed significant overrepresentation across several domains of biological processes and molecular functions (Figure 1). Most of these domains could be segmented within immune system processes, consistent with the previously described blood phenotype of FASD [7].

### 2.6. Cis-Regulatory DNAm Elements Associated with FASD

To explore the presence of potential cis-regulatory DNAm elements (CpGs regulating transcription of neighboring genes) in FASD, we first attempted to match DMRs with RNA-seq results (Appendix A); however, since there were no differentially expressed genes with an FDR < 0.05, we could not match DMRs with significant differentially expressed genes at a genome-wide significance level. Three genes—*ABR*, *GNAL*, and *SEC61G*—with nominal significant differential expression could be matched with an associated DMR (Appendix A). None of these genes were in the top 100 genes of the RNA-seq results.

Next, we employed a robust computational method to examine FASD cis-eQTMs. Median methylation levels of DMRs were correlated with log-transformed counts of associated genes. A more relaxed FDR cut-off of < 0.1 was used to select DMRs for this analysis (Appendix A). Six significant eQTMs were identified (Figure 2, Table 1 and Appendix A), comprising, in total, 21 CpGs. Five eQTMs represented a negative correlation (Table 1); in other words, a lower DNAm of DMR was associated with increased gene expression. DMRs linked to these eQTMs were hypomethylated in the FASD cohort compared to healthy controls. The gene expression of genes associated with these DMRs was increased in FASD (not at the genome-wide significance level), as expected. The eQTM with a positive correlation was associated with a hypermethylated DMR in FASD, with increased gene expression (Table 1).

## 3. Discussion

For over 50 years, FASD has been a recognized developmental disorder, but still, little is known about the exact molecular pathophysiological mechanism underlying the FASD phenotypes. Recent studies investigated patterns of aberrant DNAm in FASD and provided valuable insight into certain aspects of FASD [4,8,9]. Since epigenetic factors, such as DNAm, are strongly associated with the regulation of gene expression, we sought to explore gene expression in FASD in an attempt to further clarify FASD pathogenesis underpinnings. Here, we present the first human whole blood RNA-seq data of FASD individuals with ≥ three points in the Four-Digit score. Our results indicate a limited effect of FASD on whole blood transcriptomics. However, the combination of DNAm and gene expression data in an eQTM analysis provided a set of significant clusters of methylation probes correlated with gene expression, which present further insights into the molecular footprint of FASD in whole blood.

In the eQTM analysis, we identified six DMRs associated with changes in gene expression (*p*-value < 0.05) located near the following genes: *SEC61G*, *REEP3*, *ZNF577, HNRNPF*, *MSC*, and *SDHAF1*. This analysis allowed us to identify eQTMs mediated specifically by FASD, since eQTMs were inferred from FASD DMR associations instead of unsupervised CpG-gene expression couples. In four eQTMs (mapped to *SEC61G*, *REEP3*, *HNRNPF*, and *SDHAF1*), the gene methylation-expression correlation was apparently completely driven by the FASD cohort. The eQTMs mapped to *ZNF577* and *MSC* were seemingly correlations present in healthy population but lost in individuals with FASD. These results suggest the uncoupling of normal DNAm-gene expression associations in FASD, which calls for further investigation.

We hypothesize that the identified FASD-mediated whole blood eQTMs may serve as a combined (multi-layered omics) biomarker. Further work is required to establish the viability of such a biomarker. Two of the six genes associated with an eQTM—*HNRNPF* and *REEP3*—stand out due to their possible biological relevance in FASD, and they are discussed below.

In view of *HNRNPF*, we found a negative correlation between the DNAm level of the gene body DMR (chr10:43891459-43892075) and the expression of the gene, i.e., with lower DNAm levels, gene expression is increased. Previously, a negative association between the DNAm of the *HNRNPF* gene and alcohol consumption in whole blood has been reported, which is in line with the hypomethylated DMR we observed [10]. The *HNRNPF* (#601037 OMIM) gene encodes Heterogeneous Nuclear Ribonucleoprotein F (HNRNPF), which belongs to the family of heterogeneous nuclear ribonucleoproteins (hnRNPs). HnRNPs form complexes with heterogeneous nuclear RNA and are functionally associated with precursor–messenger RNA (pre-mRNA) splicing. The alternative splicing of genes may have drastic consequences for gene function, and since hnRNPs are ubiquitously expressed (according to https://www.proteinatlas.org/), pathogenic variants in hnRNP genes may lead to variable neurodevelopmental disorders [11]. The clinical characteristics are similar to FASD, including intellectual disability, attention deficit hyperactivity disorder, growth delay, and facial dysmorphisms. However, pathogenic variants in hnRNPs comprise loss-of-function variants, while the eQTM we observed leads to HNRNPF overexpression in FASD. *HNRNPF* pathogenic variants have previously not been associated with a clinical phenotype. In the context of the abovementioned previous findings and our results, we hypothesize that the DMR in *HNRNPF* may represent a direct functional effect driven by PAE, contributing to the manifestation of FASD during early development through the alternative splicing of multiple genes. Considering our results (in a postnatal FASD cohort) we furthermore hypothesize that, later in life, these effects remain to echo functionally in the whole blood of FASD patients. It remains to be investigated whether FASD is indeed associated with alternative splicing and if HNRNPF drives the pathophysiology of alternative splicing.

Considering *REEP3*, we also found a negative correlation between the DNAm level of the *REEP3* promoter DMR (chr10:65280473-65280961) and the expression of the gene. *REEP3* (#609348 OMIM) encodes Receptor Accessory Protein 3 (REEP3), which belongs to the Receptor Expression Enhancing Proteins (REEP) family. A recent study showed an association between the DNAm of *REEP3* at birth and periconceptional folate intake [12]. A DMR located in an intergenic region, downstream of *REEP3* (chr10:65733092-65733575), was shown to be one of the ten most responsive DMRs to folate supplementation [12]. While the DMR of the currently detected eQTM does not overlap directly, we hypothesize it could be highly relevant for FASD pathogenesis. Previously, it has been reported that chronic alcohol consumption was associated with an impaired folate absorption [13]. Furthermore, adverse early development outcomes in newborns with PAE have been linked to defective folate absorption [14,15]. Therefore, it is plausible that the *REEP3* eQTM represents an intersection of a “double hit” effect of PAE, i.e., the adverse effects of PAE plus PAE-induced impaired folate absorption. This is an important issue for future research in the context of FASD prevention with folate acid supplementation [16].

Reviewing the whole blood gene expression patterns of individuals with FASD compared to healthy controls, no genes with significant (FDR < 0.05) differential expression were detected. This result is likely to be related to the type of samples used for analysis. Since FASD is mainly considered a neurodevelopmental disorder, the biggest effect on gene expression can be expected in brain tissue (during prenatal development) and not in later-life whole blood samples. Contrived models of FASD in monkeys have indeed shown severe transcriptional effects in neuronal cells [17]. Nonetheless, FASD is characterized by a blood phenotype entailing increased susceptibility to infections [7,18], so a transcriptional effect on immune cells in whole blood can be imagined. We believe we lacked the power to identify differentially expressed genes since this study was not designed to specifically explore the implications of FASD on the blood phenotype, as there was no particular selection for immune-compromised individuals with FASD. However, in dissecting the RNA-seq data in a WGCNA, we were successful at identifying a cluster of co-expressed genes significantly associated with FASD, even after correction for relative cell type distribution. The functional annotation of this cluster in a gene set overrepresentation analysis showed associations with several immune processes. The future study of (selected) individuals with FASD is needed to confirm these associations, which could also provide insight into how PAE influences immune system physiology.

For the most part, DNAm is tissue- and cell-specific and associated with tissue- and cell-specific regulatory elements. However, whole blood is frequently used as a proxy tissue to extrapolate DNAm patterns and hypotheses for phenotypes relating to other tissues. For some genomic regions, DNAm is consistent between tissues: in one study of brain, thyroid and heart DNAm patterns, 9926 genomic regions with non-tissue-specific DNAm were identified, which also correlated with gene expression across tissues [19]. Among these regions were the currently identified *SEC61G*, *REEP3*, and *MSC* genes. However, to the best of our knowledge, there is no such data for the comparison of blood and brain DNAm patterns. The results of the current study with respect to the biological inference of FASD brain/developmental pathophysiology therefore need to be interpreted with caution.

Our work presents several limitations. Hypotheses on the pathophysiology of FASD, as concluded from this study, are based on biological inference and warrant cautious interpretation. Second, as discussed above, the findings of this study may be somewhat limited due to the use of whole blood samples for eQTM analysis. Third, this study only included participants of Polish descent. Future research will need to be undertaken to confirm the eQTM associations in different populations. Fourth, although age was included as a covariate in both the DNAm and RNA-seq analysis, we cannot exclude a residual bias stemming from age differences between cases and controls. Lastly, we could only perform a cis-eQTM analysis, since there was limited power for trans-eQTM analysis (DNAm-related associations with the expression of remote genes).

The present study on DNAm and gene expression profiles within a cohort of individuals with FASD and healthy controls provided, as aforementioned, a limited number of functional molecular leads. Future studies in this context may focus, therefore, on maximizing the sample size in order to also detect low-effect functional mechanisms of FASD. Moreover, additional omics layers (e.g., genomics and metabolomics), in combination with state-of-the-art integrative algorithms, may contribute to the elucidation of underlying molecular mechanisms, improving the diagnosis and even prognosis of FASD [20]. Although challenging to obtain, PAE severity and duration, as well as the level of FASD severity (Four-Digit score), should ideally be incorporated in such analyses. Alternatively, for unraveling the molecular mechanism of FASD, future research may focus on ex vivo and in vitro FASD models. For example, human immortalized pluripotent stems cells (hiPSCs) can be challenged at virtually all stages during development. Based on such hiPSCs, the construction of complex brain organoid models may provide important leads regarding the molecular function of neurons and their interaction with other types of brain cells during early development. This concept was recently reported by Arzua et al. and Zhu et al. [21,22]. In order to study such complex models in detail, further investigations should apply state-of-the-art spatial single cell-based high-throughput sequencing of RNA.

In summary, the aim of this study was to evaluate the influence of FASD-associated DNAm changes on gene expression in whole blood and its implications for our comprehension of FASD pathology. This research is the first to assess whole blood gene expression and eQTMs in FASD, and it has set the stage for future research into multi-omics biomarkers.

## 4. Materials and Methods

### 4.1. Subject and Sample Collection

All subjects included in this study involved a subset of a previously described cohort by Cobben et al. [4]. Informed consent was obtained from all subjects and/or legal guardians. The study was approved by the medical Ethical Committees of the Wroclaw University Hospital and the Medical University of Bialystok, Poland and by the medical Ethical Committee of the Academic Medical Center, Amsterdam, The Netherlands [4]. The present cohort involved whole blood DNA and RNA samples obtained from 12 individuals diagnosed with FASD and 51 healthy individuals. The selection criteria of individuals with FASD were defined according the Four-Digit score, where participants were included if they scored ≥ three points for alcohol exposure, central nervous system dysfunction, facial abnormalities, and growth faltering.

### 4.2. Bioinformatics Analyses

Analyses were carried out with Bioconductor (v3.15.2) packages in R (v4.2).

### 4.3. DNA Methylation Profiling and Analysis

DNA was extracted from whole blood using the FlexStar (Autogen, MA, USA) instrument, according to the manufacturer’s protocol. DNA quantity and quality control and DNAm profiling were performed by GenomeScan in Leiden (ISO/IEC 17025 certified). Briefly, bisulfite-converted DNA was amplified and subsequently hybridized on the HM450K array, according to the manufacturer’s protocol. Raw data quality control was performed using MethylAid (v.1.30; default settings) [6]. Probes annotated to the allosomes, known to involve genetically polymorphic sites (minor allele frequency > 0.01), or susceptible to cross-hybridization were removed from the dataset. Next, the data were normalized using the preprocessNoob. Blood cell type estimation was performed using the method of Houseman et al. [23]. PCA (performed with the prcomp function) was performed in order to explore and identify possible confounding factors; technical and metadata were correlated with the first eight PCs. Differentially methylated positions (DMPs) were detected using minfi (v.1.42), applying the linear model function limma (lmfit, v.3.52.4) package [24,25]. DMP analysis was based on normalized β-values. Relative blood cell distribution estimations, including the CD8+, CD4+, natural killer cell, B-cell, monocyte, and granulocyte, were included as variables in the model formula. DMPs with an FDR < 0.05 were considered as genome-wide significant. Differentially methylated regions (DMRs) were detected using the DMRcate package (v.2.10) [26]. DMRs were identified using the following criteria: a minimum of two consecutively significant DMPs (FDR < 0.05) within a 1000 nucleotide range, and for integrative/eQTM analyses, a minimum of two consecutively significant DMPs (FDR < 0.1) within a 1000 nucleotide range. DMRs were ranked based on Stouffer’s coefficient (SC). DMRs with an SC < 0.05 were considered as genome-wide significant [26].

### 4.4. RNA Sequencing and Analysis

RNA was isolated from PAXgene whole blood specimens. RNA was extracted using the RNA mini kit (cat# 763134) (Qiagen, Hilden, Germany), conforming to the manufacturer’s protocol. RIN values were obtained, conforming to the manufacturer’s protocol, using a Fragment analyzer 5200 (Agilent Technologies, Santa Clara, CA, USA). To obtain a gene expression profile, libraries of complementary DNA (cDNA) to messenger RNA (mRNA) were generated (NEBNext Ultra II Directional RNA Library Prep Kit for Illumina, NEB #E7760S/L). Generated libraries were (SE50) sequenced using the NovaSeq 6000 (Illumina, San Diego, CA, USA). Image analysis and base-calling were carried out using an Illumina data analysis pipeline involving Real-Time Analysis (RTA) and bcl2fastq (v2.20). The quality of the data was assessed with FastQC (v0.11.9) [27]. Alignment to the hg38 human genome was performed using STAR (v2.7.3) with the default parameters [28]. SAMtools (v1.11) and featureCounts (v2.0.1) were used for the alignment and count quantification [29,30]. Additional QC metrics were generated using MultiQC (v1.9) to check samples with an RIN < 7.0 manually. PCA (performed with the prcomp function) was employed for the analysis of relationships between samples and outlier detection. The analysis of differential gene expression was performed using DESeq2 (v1.36.0) [31]. The abovementioned relative blood cell distributions (inferred from DNAm data) were included as covariates in the model formula.

### 4.5. Weighted Correlation Network Analysis

WGCNA was carried out with the WGCNA package (v1.71) [32]. Variance stabilizing transformation (VST) normalized gene counts of genes with a non-zero count in at least 90% of samples were used as the input for WGCNA to prevent the noise of lowly expressed genes. The sources of variation (age, gender, and blood cell type distribution) were corrected for with the removeBatchEffect function of limma (v3.46.0) [24]. Based on the network topology of the data, we chose a soft thresholding power β of 6, to which co-expression similarity was raised to calculate adjacency. Then, the standard Pearson correlation method was used to construct signed networks (modules) of co-expressed genes. The correlation among the FASD phenotype and identified modules was studied using module eigengene networks. We considered correlations with *p*-values < 0.05 significant.

We performed gene set ORA with the clusterProfiler package (v4.6.0) [33], employing the default settings, using all WGCNA input genes as the background, and testing for GO terms. We considered terms with an FDR < 0.05 significantly overrepresented. The treeplot function was used for hierarchical clustering based on pairwise similarities of the overrepresented terms.

### 4.6. Expression Quantitiative Trait Mehtylation (eQTM) Analysis

eQTM analysis was performed as previously described [34]. Briefly, for each DMR in each sample, the median methylation level was calculated. Next, Pearson’s correlation coefficient was calculated between the DMR methylation medians and log-transformed counts of the associated gene. The 95% confidence intervals were calculated based on 100,000 bootstraps. *p*-values were obtained by comparing correlation coefficients with a null distribution using a resampling approach. DMR–gene correlations with *p* < 0.05 were considered significant. The results of the integrative methylation expression analysis were reported according to GRCh37 (hg19).

## Figures and Tables

**Figure 1 ijms-24-06601-f001:**
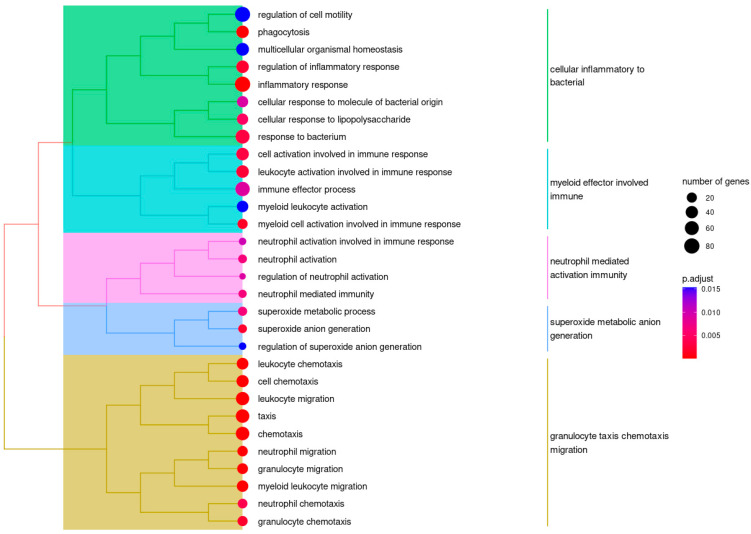
Hierarchical clustering of significantly overrepresented GO terms in module brown. Semantic cluster tags are given to the right of each cluster. Most GO terms are related to immune processes. We believe this signifies immune system abnormalities and is a reflection of the blood phenotype of FASD, which is characterized by an increased risk of infections.

**Figure 2 ijms-24-06601-f002:**
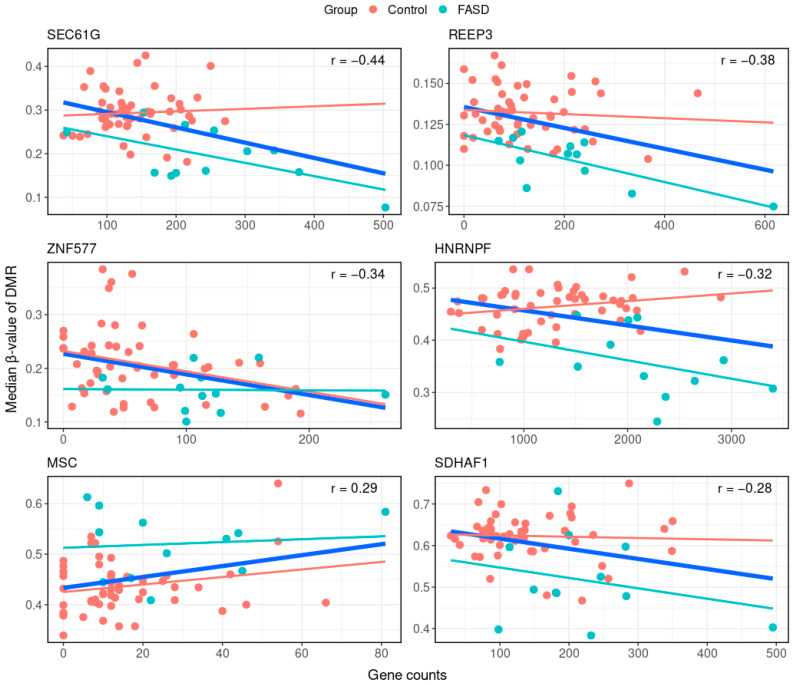
eQTMs. Top six significant DMR–gene expression eQTMs associated with FASD. *x*-axes represent gene counts, *y*-axes represent median β-values of the eQTM-associated DMR. Each sample is represented by a dot, where blue dots are samples of individuals with FASD and red dots are samples of healthy controls. Relationships between DMR methylation and gene expression are given by linear correlation lines. The thick blue line represents the eQTM from the combined analysis of FASD and control data, for which Pearson’s correlation coefficient is also given. Thin lines are group-specific correlations.

**Table 1 ijms-24-06601-t001:** DMR–gene expression eQTMs.

Position	Gene	Correlation Coefficient	*p* Value	DMR Feature	DMR Direction	Gene Expression Direction
chr7:54827528-54827677	*SEC61G*	−0.44	2.29 × 10^−3^	Promoter	Hypo-methylated	Overexpression
chr10:65280473-65280961	*REEP3*	−0.38	3.69 × 10^−3^	Promoter	Hypo-methylated	Overexpression
chr19:52391078-52391090	*ZNF577*	−0.34	8.33 × 10^−3^	1st exon	Hypo-methylated	Overexpression
chr10:43891459-43892075	*HNRNPF*	−0.32	1.55 × 10^−2^	Gene body	Hypo-methylated	Overexpression
chr8:72758461-72758701	*MSC*	0.29	1.98 × 10^−2^	Promoter	Hyper-methylated	Overexpression
chr19:36484731-36485360	*SDHAF1*	−0.28	3.12 × 10^−2^	Promoter	Hypo-methylated	Overexpression

## Data Availability

Summary statistics supporting the results in this manuscript have been provided as Appendix A. Raw gene expression sequencing data (.fastq) of the results presented in this manuscript are available on reasonable request and under a data transfer agreement (DTA) following the Dutch data protection act (DPA) at the European Genome–Phenome Archive: https://ega-archive.org/studies, under accession identifier EGAS00001006899. Raw DNA-methylation profiles (.idat) and metadata are freely available at the NCBI GeneExpression Omnibus (GEO, https://www.ncbi.nlm.nih.gov/geo), under accession identifier GSE112987. All scripts used for statistical analyses are described in Appendix A.

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
