# Peer review of "Expression Quantitative Trait Methylation Analysis Identifies Whole Blood Molecular Footprint in Fetal Alcohol Spectrum Disorder (FASD)"

_ijms, 2023, doi:10.3390/ijms24076601_

Round 1

Reviewer 1 Report

This article was very difficult to read because of the large amount of factual and numerical information. Also, for me, as someone who does not work with molecular biology on a regular basis, I would have liked more general explanations, since it is not a fact that the reader will be a molecular biologist. However, I can't help but note the high level of the article, its practical applicability, and the very high quality explanation of the results obtained. All the questions I had while reading the article were answered literally in the following paragraphs.

Author Response

Reviewer 1 comments to Author

This article was very difficult to read because of the large amount of factual and numerical information. Also, for me, as someone who does not work with molecular biology on a regular basis, I would have liked more general explanations, since it is not a fact that the reader will be a molecular biologist. However, I can't help but note the high level of the article, its practical applicability, and the very high quality explanation of the results obtained. All the questions I had while reading the article were answered literally in the following paragraphs.

Reply: We thank the Reviewer for their nice comments. We have amended the text and tried to include more general explanations to appeal to a broader audience.

Reviewer 2 Report

Krzyzewska et al. perform gene expression profiling studies and identified 6 differentially methylated regions associated with changes in gene expression in FASD. However, there are several points that need to be addressed to provide more support to their claims.

Major points:

1.    Have the authors tried to perform comparative studies with individuals who are not of Polish descent? How might the demographics of the dataset impact these studies?

2.    How does the difference in average age of control and FAS individuals impact the study?

3.    For lines 103-111, provide the analysis method, data and relevant citations.

4.    Add the principal component analysis mentioned in line 126 as a supplementary figure.

5.    Have the authors analyzed the GO terms in module turquoise (0.19) and module pink (-0.19)? It would be interesting to see what each of the modules represent and how they might impact FASD.

6.    The authors mention the use of HNRNPF and REEP3 as potential biomarkers of FASD. Can the authors perform validation experiments to measure their gene expression in FASD samples that were not selected for this study as mentioned in the methods?

7.    Can the authors build a predictive model for FASD based on the weighted expression of the identified 6 genes? What would the predictive power of such a model be?

Minor Points:

1.    Elaborate on HM450K and MethylAid quality control checks in line 90-91 and/or provide a citation.

2.    Describe how the principal component analysis was performed in the Methods section.

3.    In Supplementary figure 1, spell out the cell types or describe the shorthand notations used.

4.    Label the subfigures as A,B,C in Supplementary figure 1 and reference them appropriately in the main text.

5.    Expand FDR.

6.    Line 320 needs a citation.

7.    Grammatical errors, typos, and punctuation should be thoroughly checked in the entire manuscript. 

Author Response

Reviewer 2 comments to Author

Krzyzewska et al. perform gene expression profiling studies and identified 6 differentially methylated regions associated with changes in gene expression in FASD. However, there are several points that need to be addressed to provide more support to their claims.

Major points

  1. Have the authors tried to perform comparative studies with individuals who are not of Polish descent? How might the demographics of the dataset impact these studies?

Reply: The Reviewer raises an important point. In the primary paper on this manuscript (doi: 10.2217/epi-2018-0221), DNAm findings were validated in other cohorts. Sadly, for the current experiment, RNA-seq data was only available for participants of Polish descent. Therefore, findings could not be validated in cohorts of different descent. Future research will need to be undertaken to confirm the eQTM associations in different populations. We have now addressed this in the discussion (line 280), thank you for pointing this out.

  1. How does the difference in average age of control and FAS individuals impact the study?

Reply: We acknowledge that age can be considered as confounder, in particular in DNAm studies. However, we assumed that the mean age difference between cases and controls was limited and therefore inclusion as covariate sufficiently addresses such bias. Still, we cannot exclude residual bias. We have addressed that in the limitations now (line 282).

  1. For lines 103-111, provide the analysis method, data and relevant citations.

Reply: we agree with the reviewer that lines 103-111 needed some clarification regarding methodology, data and relevant citations, we therefore amended this section.

  1. Add the principal component analysis mentioned in line 126 as a supplementary figure.

Reply: We thank the Reviewer for this suggestion. We have added the PCA results as a figure. Also, we have added the PCA results of DNAm analysis as a figure.

  1. Have the authors analyzed the GO terms in module turquoise (0.19) and module pink (-0.19)? It would be interesting to see what each of the modules represent and how they might impact FASD.

Reply: With all respect to the Reviewer, we feel that exploring these modules would be too speculative, since the nominal p-values of these modules were both 0.1. We therefore chose to not further analyse these modules.

  1. The authors mention the use of HNRNPF and REEP3 as potential biomarkers of FASD. Can the authors perform validation experiments to measure their gene expression in FASD samples that were not selected for this study as mentioned in the methods?

Reply: This is an excellent idea for a follow-up study. However, we sadly do not have access to RNA-seq data of participants that were not selected.

  1. Can the authors build a predictive model for FASD based on the weighted expression of the identified 6 genes? What would the predictive power of such a model be?

Reply: We also feel that this could be an excellent idea for a follow-up study and thank the Reviewer for this excellent suggestion. However, the focus of this manuscript was to investigate eQTM associations of FASD. Thus, we feel that this goes beyond the scope of the manuscript.

Minor Points

  1. Elaborate on HM450K and MethylAid quality control checks in line 90-91 and/or provide a citation.

Reply: These are the default settings. We have amended the text and provided the citation.

  1. Describe how the principal component analysis was performed in the Methods section.

Reply: We thank the Reviewer for this suggestion. We have added a section to the methods (line 348).

  1. In Supplementary figure 1, spell out the cell types or describe the shorthand notations used.

Reply: We have spelled out the cell types in the text explaining the Supplementary Materials (line 381).

  1. Label the subfigures as A,B,C in Supplementary figure 1 and reference them appropriately in the main text.

Reply: We thank the Reviewer for this nice suggestion. We have labeled the panels in supplementary figure 1 and reference to them in the text accordingly.

  1. Expand FDR.

Reply: FDR is now expanded at first mention in the text (line 134).

  1. Line 320 needs a citation.

Reply: We have referenced to the DMRcate citation concerning the use of Stouffer’s coefficient.

  1. Grammatical errors, typos, and punctuation should be thoroughly checked in the entire manuscript.

Reply: The manuscript has been proof-read by a native speaker, and the manuscript has been amended where needed.

Round 2

Reviewer 2 Report

The authors have addressed most of the comments and there are a few points to be addressed before publication:

1) Supplementary Tables are missing.

2) In line 130, the authors need to describe the inputs used for PCA. They mention no outliers but there are some points that do not cluster together with the rest. Describe the discrepancy.  

3) Include a section in Discussion about future studies that can be performed and the impact of this manuscript. 

Author Response

Dear Editor,

We are grateful to you and the reviewers for again carefully reading and assessing our manuscript. A point-by-point reaction to the comments is provided below. All changes have been highlighted in the revised version of the manuscript.

We are pleased to re-submit the revised manuscript to IJMS and hope it will now be suitable for publication.

Sincerely,

Peter Lauffer

Reviewer 1 comments to Author

1) Supplementary Tables are missing.

We thank the Reviewer for pointing this out and apologize for the mistake. We have included the Supplementary Tables in the zip file of supplementary materials.

2) In line 130, the authors need to describe the inputs used for PCA. They mention no outliers but there are some points that do not cluster together with the rest. Describe the discrepancy. 

We have added the inputs used for PCA as recommended. We have used filtered gene counts.

As for the PCA results, we have clarified our interpretation of the results. For QC we have used FastQC. Therefore, we have omitted the word “outliers” from the explanation of PCA results, since this suggests we used PCA for QC. We have used PCA as an explorative analysis to illuminate the structure of our data. Our main conclusion from the PCA is that there are no samples that unexpectedly cluster separately from the rest of the samples. We thank the Reviewer for reflecting on this.

3) Include a section in Discussion about future studies that can be performed and the impact of this manuscript.

We thank the Reviewer for this nice suggestion. We have added a section at the end of the discussion that discusses the impact of the current study, and future studies that might contribute to elucidating FASD pathogenesis.